# GardeninA confers neuroprotection against environmental toxin in a *Drosophila* model of Parkinson's disease

Urmila Maitra[1✉], Thomas Harding[1], Qiaoli Liang[2] & Lukasz Ciesla [1✉]

Parkinson's disease is an age-associated neurodegenerative disorder characterized by the progressive loss of dopaminergic neurons from the midbrain. Epidemiological studies have implicated exposures to environmental toxins like the herbicide paraquat as major contributors to Parkinson's disease etiology in both mammalian and invertebrate models. We have employed a paraquat-induced Parkinson's disease model in *Drosophila* as an inexpensive in vivo platform to screen therapeutics from natural products. We have identified the polymethoxyflavonoid, GardeninA, with neuroprotective potential against paraquat-induced parkinsonian symptoms involving reduced survival, mobility defects, and loss of dopaminergic neurons. GardeninA-mediated neuroprotection is not solely dependent on its antioxidant activities but also involves modulation of the neuroinflammatory and cellular death responses. Furthermore, we have successfully shown GardeninA bioavailability in the fly heads after oral administration using ultra-performance liquid chromatography and mass spectrometry. Our findings reveal a molecular mechanistic insight into GardeninA-mediated neuroprotection against environmental toxin-induced Parkinson's disease pathogenesis for novel therapeutic intervention.

[1] Department of Biological Sciences, University of Alabama, 2320 Science and Engineering Complex, Tuscaloosa, AL 35487-0344, USA. [2] Mass Spectrometry Facility, Department of Chemistry and Biochemistry, University of Alabama, 2004 Shelby Hall, Tuscaloosa, AL 35487-0336, USA. ✉email: umaitra@ua.edu; lmciesla@ua.edu

Neurodegenerative disorders, including Parkinson's (PD) and Alzheimer's disease (AD) have emerged as one of the greatest medical challenges of the 21st century[1,2]. PD is the second most common age-associated neurodegenerative movement disorder primarily characterized by the progressive loss of dopaminergic neurons from the midbrain[3]. Neurodegenerative diseases (NDs) cost the U.S. economy billions of dollars each year in direct health care costs, and the costs to the society will further increase. Unfortunately, the pathogenesis of idiopathic PD is not well understood, and current therapies only provide symptomatic relief but fail to slow down or prevent disease progression. A dire need exists to develop novel therapies targeting early molecular pathways involved in the development and progression of PD.

In addition to genetic mutations, numerous epidemiological studies have strongly indicated that occupational exposure to environmental toxins, such as pesticides like paraquat (PQ), rotenone, or neurotoxic metals, is associated with a higher risk of developing idiopathic PD[4,5]. Exposure to PQ is known to induce oxidative stress and mimic clinical signs similar to PD, including reduced motor function and selective dopamine neuron degeneration, which can be rescued by administration of Levodopa (L-dopa) in both mammalian and *Drosophila* models[6,7]. A recent study has demonstrated the link between PQ exposure and PD using human stem cells[8]. Moreover, our recently published data using transcriptomic analysis identified genes in *Drosophila* PD model, whose mammalian orthologs are implicated in PD pathogenesis[6]. PQ has also been shown to induce dysregulated inflammatory responses related to neuronal cell death[6]. Both oxidative stress and neuroinflammation have been associated with PD pathogenesis in both mammalian and *Drosophila* models[9]. These similarities strongly underscore the utilization of the *Drosophila* PD model as an in vivo screening platform to identify novel plant-derived therapeutics due to the presence of evolutionarily conserved mechanistic pathways that are associated with the disease pathogenesis[10].

Numerous epidemiologic studies have shown a plant-based diet, especially rich in flavonoids is associated with lower mortality and decreased risk of developing age-associated chronic diseases[11–13]. Plants produce specialized metabolites, which provide protection from environmental stressors both in plants and animals feeding on them[13]. However, the exact neuroprotective mechanism of these phytochemicals is not clear. For many years, it has been promulgated that these molecules work as free radical scavengers, preventing DNA damage, especially in the mitochondria by direct antioxidant activity[13,14]. Rigorous experiments involving human clinical trials disproved this theory[13]. However, the antioxidant theory is still alive in our society, a belief that fuels $3 billion global industry selling pills and drinks with "natural antioxidants"[15]. The emerging evidence suggests that phytochemicals may exert disease-preventive and therapeutic actions by interacting with certain evolutionarily conserved cellular response pathways, as suggested by neurohormesis and xenohormesis hypotheses[14,16].

Plant-derived molecules have a long history of use as templates for the development of novel medicines for numerous diseases, including NDs[17]. Levodopa (L-dopa), currently the most commonly prescribed drug for PD, was first identified from the broad bean (*Vicia faba*)[18]. *Mucuna pruriens* has been considered as a possible alternative in the management of PD in patients who cannot afford L-dopa therapy[19]. Molecules of dopamine receptor agonists also used in PD therapy, pergolide and lisuride are synthetic derivatives of ergoline, a natural alkaloid first described in ergot[20]. Several natural compounds produced by plants and microbes have been recently found to extend lifespan and provide neuroprotective effects, for example, rapamycin, fisetin,

curcumin, and its derivatives, quercetin and its derivatives, 7,8-dihydroxyflavone[21–24]. Recently, Zhang et al.[25] showed that a combination of the chemotherapeutic drug, dasatinib, and a natural flavonoid, quercetin, alleviates Aβ-associated senescence and improves cognitive deficits in a mice model of AD. Other examples of compounds that are protective against toxin-induced neurodegeneration include metformin[26], and catalpol[27]. However, most studies related to flavonoids solely concentrate on the antioxidant theory and lack in-depth molecular mechanisms of action underlying the neuroprotective phenotypes. In addition, several studies focus on evaluating the neuroprotective effects in cell culture models using super-physiological concentrations of phytochemicals, which mostly translate to poor clinical outcomes[28]. Another concern regarding the use of natural products, especially flavonoids as therapeutics is their poor bioavailability[29]. There is a need to evaluate the neuroprotective effects of natural products at pharmacologically relevant concentrations in the context of a living organism.

Here, we have employed a well-established PQ-induced PD model in *Drosophila* as an in vivo screening platform to identify neuroprotective natural compounds. Plants produce various molecules, many of which interact with evolutionarily conserved pathways, as these compounds evolved either in common ancestor or in response to insect herbivory. This makes the *Drosophila* PD model an appropriate and inexpensive screening platform during the initial stages of the drug discovery pipeline before validations in mammalian models. We have identified a flavonoid, GardeninA, whose neuroprotective mechanism against PQ-induced PD symptoms is not solely dependent on its antioxidant activities. The polymethoxyflavone, GardeninA is found in the gum of the medicinal plant, *Gardenia resinifera* Roth. Another recent study revealed significant anti-depressant actions of GA, thereby suggesting neuropharmacological effects in mice[30]. However, this is the first study to reveal the neuroprotective potential of GardeninA against environmental toxin-induced neurodegenerative phenotypes in *Drosophila*. GardeninA-mediated protection involves modulation of multiple pathways, including the neuroinflammatory and cellular death responses that are intricately associated with PD pathogenesis. Furthermore, this is the first report showing the bioavailability of the neuroprotective flavonoid, GardeninA in *Drosophila* heads at pharmacologically relevant nanomolar doses using ultra-performance liquid chromatography and mass spectrometry.

## Results

**The flavonoid GardeninA confers protection against PQ-induced toxicity and mobility defects in *Drosophila*.** We employed an environmental toxin-induced model in *Drosophila*, that recapitulates the behavioral and pathological symptoms of PD, as a screening platform to identify phytochemicals with therapeutic potential against the herbicide PQ toxicity using the feeding regimen shown in Fig. 1a. We have previously shown that exposure to PQ significantly reduces survival and impairs climbing abilities in the wild-type *Drosophila* strain, *Canton S*[6,31]. We have only used adult male flies due to their higher sensitivity to PQ toxicity leading to neurodegenerative PD phenotypes at an earlier stage than female flies, consistent with a higher prevalence of PD in human male patients[31]. In addition, the feeding patterns of female flies differ based on the mating status[32]. Consistent with earlier findings, co-feeding levodopa (L-dopa) 0.5 mM, a dopamine replacement drug used as a gold standard in PD significantly increased survival and improved mobility defects after exposure to PQ (Fig. S1a, b), thereby mirroring the therapeutic effects observed in human PD patients. This underscores the suitability of this environmental toxin-induced model as a

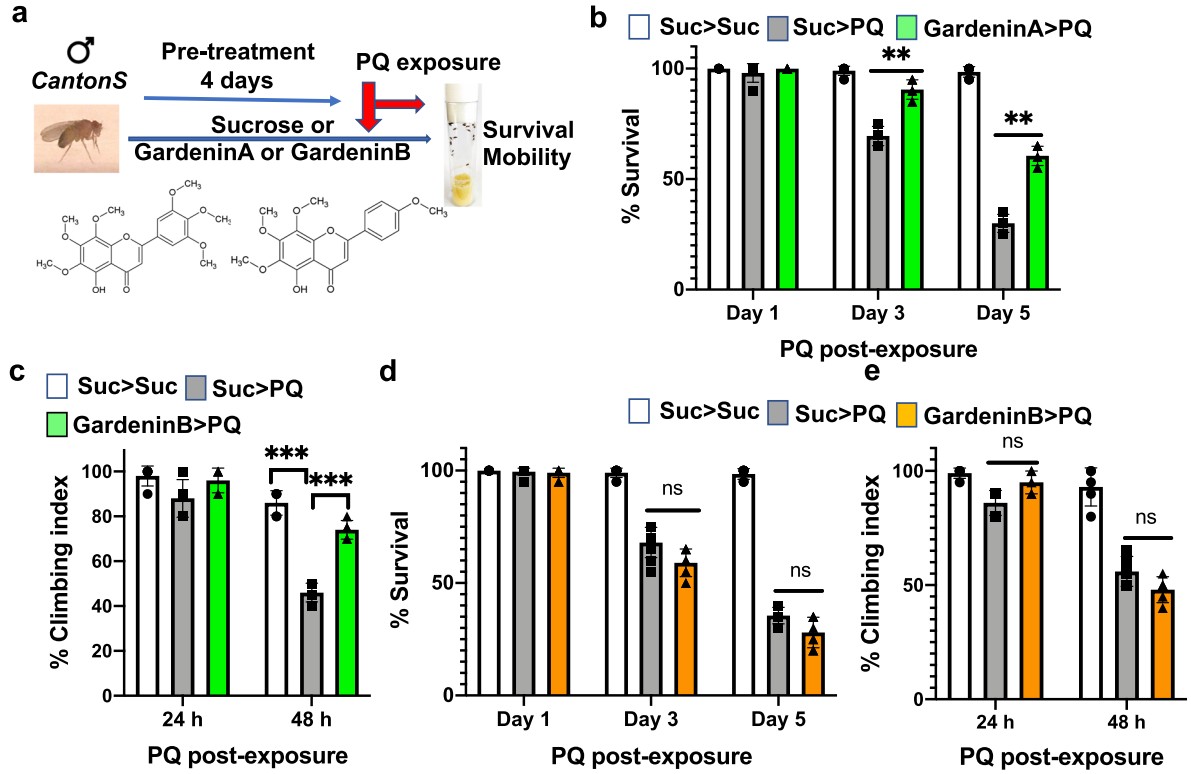

**Fig. 1 The protective effects of the flavonoid, GardeninA against paraquat-induced toxicity and mobility defects. a** Schematic representation of the feeding regimen. Adult male flies were either pre-fed for 4 days with 2.5% sucrose or GardeninA or GardeninB (10 µM) followed by continuous exposure to 5 mM PQ (white bar: Suc>Suc, gray bar: Suc>PQ, green bar: GardeninA>PQ) and were scored daily for survival and mobility abilities. **b** Survival assays were performed using adult male flies as outlined above and the number of live flies was recorded every 24 h until all of the flies died and the survival percentages were plotted at the indicated time points (Days 1, 3, and 5). Data are representative of ten independent biological replicates with ten flies per feeding conditions. Data shown represents mean ± SEM. **$p < 0.01$ based on the Mann–Whitney $U$ test. **c** The protective effect of GardeninA pretreatment on the climbing abilities of flies exposed to PQ was determined by negative geotaxis assays. The number of flies able to cross 5 cm from the bottom of the vial within 20 s were recorded and plotted at the specified time points (24 and 48 h). Data are representative of at least five independent experiments with ten flies per group. ***$p < 0.001$ based on one-way ANOVA between indicated feeding conditions. **d** Adult male flies were either pre-fed for 4 days with 2.5% sucrose or GardeninB (10 µM) followed by continuous exposure to 5 mM PQ (white bar: Suc>Suc, gray bar: Suc>PQ, orange bar: GardeninB>PQ) and were scored daily for survival assays. The survival percentages were plotted at the indicated time points (Days 1, 3, and 5). Data are representative of ten independent biological replicates with ten male flies per feeding conditions. Data shown represent mean ± SEM. **$p < 0.01$ based on the Mann–Whitney $U$ test. **e** Negative geotaxis assays were used to determine the effect of GardeninB pretreatment on the climbing abilities of flies exposed to PQ. The number of flies able to cross 5 cm within 20 s were recorded and plotted at the specified time points (24 and 48 h). Data are representative of at least five independent experiments with ten male flies per group. ***$p < 0.001$ based on one-way ANOVA between indicated feeding conditions.

screening platform to identify plant-derived therapeutics against PD pathogenesis.

Adult male flies were pre-fed for 4 days with either 2.5% sucrose solution as control or GardeninA (GA) 10 µM diluted in 2.5% sucrose solution on filter paper through ingestion, followed by exposure to 5 mM PQ continuously to induce parkinsonian symptoms until all of the flies were dead. Survival was scored every 24 h postexposure and the survival percentages on days 1, 3, and 5 post PQ exposure between different feeding groups were plotted in Fig. 1b. The data show that PQ exposure leads to a gradual increase in mortality of flies with average survival of 70% and 30% on days 3 and 5, respectively. Interestingly, GA pretreatment provided significant protection against PQ-induced toxicity with 60% survival of flies on day 5 as compared to 30% of sucrose-fed control flies ($p < 0.01$). We screened different concentrations and durations of GA pretreatments (5, 10, 20, and 40 µM) and selected four days of pre-feeding GA at 10 µM for further studies that yielded highly reproducible maximal protection against PQ toxicity in the survival assays. We also performed feeding assays using the blue food dye (1% FD&C Blue#1) to show that pretreatment with either GardeninA

or GardeninB does not lead to significant differences in the food intake abilities as compared to the sucrose-fed controls (Fig. S2).

Next, we examined whether GA was able to restore mobility defects induced by PQ using negative geotaxis assay. The flies were pre-fed for 4 days with either sucrose or 10 µM GA followed by continuous exposure to PQ and climbing assessment was performed at 24 and 48 h time points postexposure. As shown in Fig. 1c, PQ exposure significantly reduced climbing abilities, whereas GA pretreatment significantly restored PQ-induced mobility impairment to 75% as compared to 45% of the control sucrose-fed group ($p < 0.001$) at 48 h postexposure to PQ. Our next goal was to determine the effect of GardeninB (GB), another flavonoid structurally related to GA, on PQ-mediated toxicity using survival assays. GB contains fewer methoxy groups in the B ring of the flavone skeleton as compared to GA (Fig. 1a). We used the same feeding regimen as depicted in Fig. 1a, where adult male flies were pre-fed for 4 days with either sucrose or 10 µM GB diluted in 2.5% sucrose solution, and then continuously exposed to 5 mM PQ and survival was scored every 24 h. The survival percentages were plotted for days 1, 3, and 5 post PQ exposure. Surprisingly, GB pretreatment was not effective in conferring

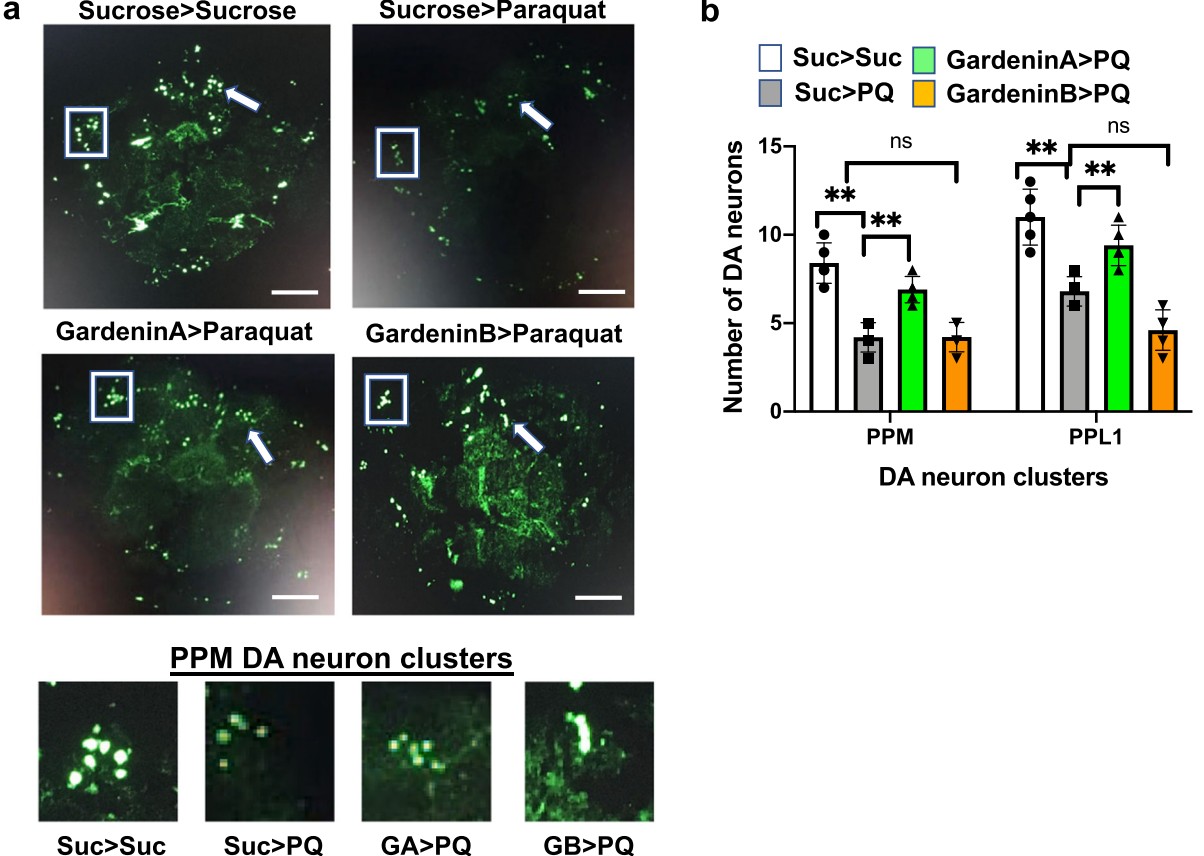

**Fig. 2 GardeninA but not GardeninB is protective against PQ-mediated loss of dopaminergic neurons. a** Confocal imaging was used to detect specific clusters of dopaminergic neurons in response to the indicated feeding conditions. Adult male flies were either pre-fed for 4 days with 2.5% sucrose or GardeninA or Gardenin B (10 μM) followed by continuous exposure to 5 mM PQ for 48 h (white bar: Suc>Suc, gray bar: Suc>PQ, green bar: GardeninA>PQ, orange bar: GardeninB>PQ). The specific DA neuron clusters in the *Drosophila* brain are depicted by the GFP-positive green signals. White boxes indicate the DA neuron protocerebral posterior lateral 1 (PPL1) cluster and the white arrow specify the protocerebral posterior medial (PPM) DA neuron cluster. Scale bars 100 μm. The bottom panel shows the enlarged representative views of the PPM DA clusters in response to the specified feeding conditions. **b** The average number of dopaminergic neurons per cluster (PPM and PPL1) in response to the indicated feeding conditions are plotted. Data are representative of at least five independent biological replicates. Data shown represent mean ± SEM. **$p < 0.01$; ns not significant based on one-way ANOVA between different feeding conditions.

protection against PQ-induced toxicity in survival assays (Fig. 1d). Next, we determined the effect of GB on PQ-induced mobility defects using negative geotaxis assay. As shown in Fig. 1e, PQ exposure significantly reduced climbing abilities and GB pretreatment failed to rescue PQ-induced mobility impairment as compared to the control sucrose-fed group. Our data confirm that the structurally related polymethoxyflavonoid, GB is not protective against PQ-induced toxicity and mobility defects. Overall, GA appears to be a potential neuroprotective candidate against PQ toxicity in a *Drosophila* PD model.

**GardeninA but not GardeninB rescues against PQ-mediated loss of dopaminergic neurons.** PQ has been shown to induce the loss of specific clusters of dopaminergic neurons (DA) in both mammalian and *Drosophila* PD models[6,31]. Since GA pretreatment improved PQ-induced mobility defects, we analyzed the effect of GA and GB on specific clusters of DA neurons in the adult *Drosophila* brain. We monitored the DA clusters using green fluorescent protein (GFP) expression driven by the tyrosine hydroxylase (TH)-promoter in *TH-Gal4; UAS-GFP* transgenic adults. Consistent with earlier findings, the numbers of DA neurons in the protocerebral posterior median (PPM) and protocerebral posterior lateral (PPL1) clusters in the posterior region of the brain were significantly lower in the PQ-fed group (PPM:

4.2; PPL1: 6.8) as compared to the control sucrose-fed flies (PPM: 8.4; PPL1: 11; $p < 0.01$) (Fig. 2a, b). However, pretreatment with GA resulted in the rescue of specific clusters of DA neurons (PPM: 6.9; PPL1: 9.4; $p < 0.01$) that were more susceptible to PQ exposure ($p < 0.01$). The lower panel highlights the enlarged version of the PPM DA clusters for different feeding conditions (Fig. 2a). Interestingly, the structurally related flavonoid, GB pretreatment failed to improve DA neuron degeneration in response to PQ (PPM: 4; PPL1: 4.8; $p > 0.05$) (Fig. 2a, b). These results suggest that GA but not GB is protective against PQ-induced neurotoxicity involving DA neuron loss.

**The antioxidant activity is not sufficient to confer neuroprotection against PQ-induced oxidative stress.** Previous studies have underscored the role of oxidative stress in the onset and progression of NDs, including AD and PD[33]. The herbicide PQ is known to induce oxidative stress leading to the clinical signs of PD in both mammalian and invertebrate models. Most of the reported studies on flavonoids claim that they provide neuroprotection by scavenging free radicals like reactive oxygen species and inducing the antioxidant response within the cells to suppress oxidative stress[34]. Therefore, to evaluate the effects of GA and GB on PQ-induced oxidative stress, the flies were pre-fed with either sucrose or GA or GB for 4 days followed by PQ exposure for 18 h

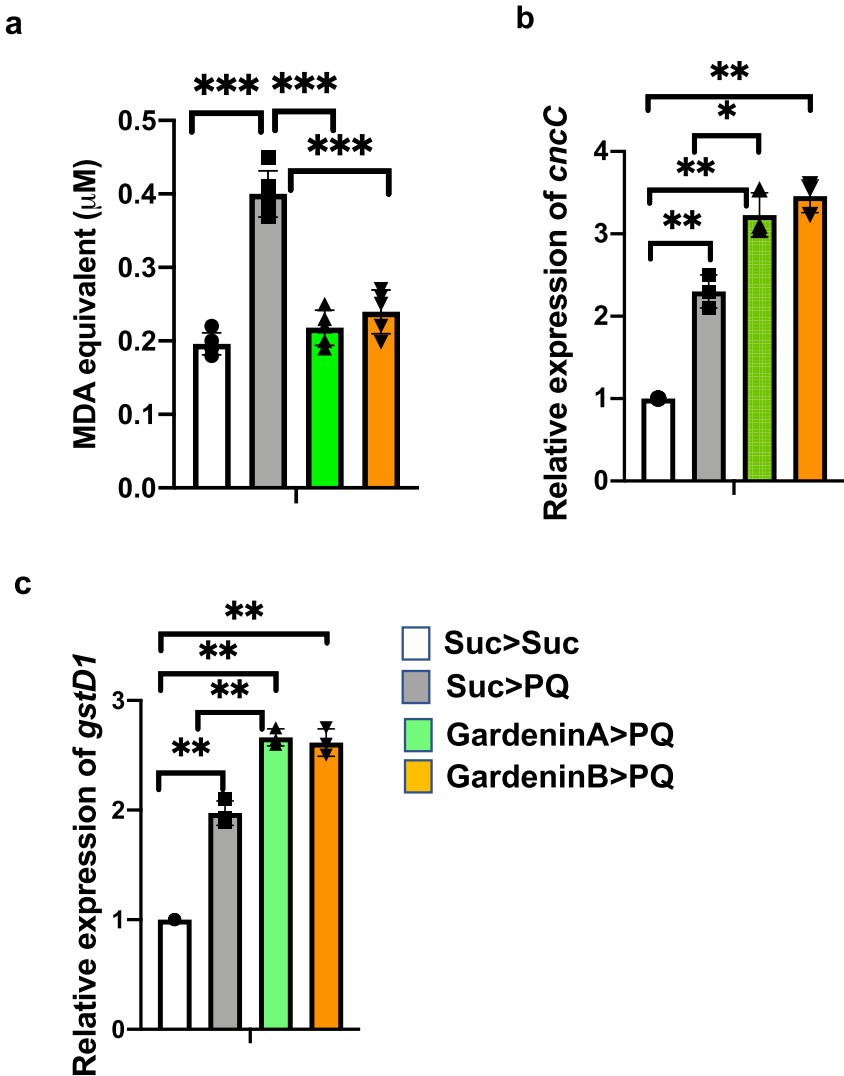

**Fig. 3 The neuroprotective effect of GardeninA against PQ toxicity is not dependent on its antioxidant activities. a** Effects of GA and GB on the levels of MDA as a marker for lipid peroxidation due to oxidative stress using a fluorometric assay. Adult male flies were either pre-fed for 4 days with 2.5% sucrose or GA or GB (10 μM) followed by exposure to sucrose or 5 mM PQ for 18 h (white bar: Suc>Suc, gray bar: Suc>PQ, green bar: GardeninA>PQ, orange bar: GardeninB>PQ). The fly head homogenates were used to detect the levels of MDA (μM) based on the calibration curve generated using the standards provided with the fluorometric kit. All experiments were performed in triplicate and represent at least three independent biological repeats. Data shown represent mean ± SEM. ***$p < 0.001$ based on one-way ANOVA between different feeding conditions. **b**, **c** Effects of GA and GB on the transcript levels of *cncC*, the human Nrf2 orthologue in *Drosophila* and the downstream oxidative stress response gene, *gstD1* using qRT-PCR. RNA was isolated from the heads of adult flies following exposure to the indicated feeding conditions. The transcript levels of *cncC* and *gstD1* were analyzed and plotted after normalization with *rp49* levels as the internal control. Each data point represents mean ± SEM. The mRNA fold changes are normalized to the sucrose-fed (Suc > Suc) flies (assigned a value of 1). **$p < 0.01$ between different feeding conditions based on the Mann–Whitney *U* test.

as it gave reproducible data between independent biological replicates at this time point. Lipid peroxidation assay was used to assess the levels of oxidative stress as a measure of malondialdehyde (MDA), one of the final products of polyunsaturated fatty acids peroxidation in the cells. As shown in Fig. 3a, PQ significantly induced oxidative stress by increasing the MDA levels (0.39 μM) relative to the sucrose-fed control flies (0.19 μM; $p < 0.001$). Interestingly, both GA and GB pretreatment significantly decreased the levels of MDA to 0.22 and 0.25 μM, respectively, and thereby reducing PQ-induced oxidative stress.

Another way of controlling oxidative stress involves the nuclear factor erythroid 2-related factor 2 (Nrf2), which acts as a regulator of cellular resistance to oxidants. Nrf2 is known to regulate the expression of a wide array of antioxidant genes and detoxifying proteins such as thioredoxins, glutathione synthetase, and glutathione S-transferases[35]. Recent studies have shown that activation of Nrf2 is protective against genetic and environmental toxin-induced neurodegenerative phenotypes, thereby highlighting its neuroprotective potential in both mammalian and invertebrate PD models. Interestingly, Nrf2 homologs are conserved in *Drosophila* and it belongs to the *cap'n'collar (cnc)* subfamily of leucine zippers, named after the *cnc* gene of *Drosophila*. We evaluated the effect of PQ exposure on Nrf2 transcript levels by quantitative real-time polymerase chain reaction (qRT-PCR) and confirmed the induction of Nrf2 mRNA to 2.3-fold compared to the sucrose-fed control group (Fig. 3b). Moreover, both GA and GB pretreatment resulted in further significant induction of the Nrf2 transcript levels, thereby supporting its role in activating the antioxidant response to

counteract against PQ toxicity. To assess the antioxidative response of Nrf2 induction, we further determined the transcript levels of an oxidative stress response gene, glutathione-S-transferase (gstD1), which encodes a detoxification enzyme downstream of Nrf2. As shown in Fig. 3c, PQ exposure led to the induction of *gstD1* transcript levels to 1.96-fold compared to the sucrose-fed control group. Our data revealed that both GA and GB induce the expression of *gstD1*, thereby suggesting the activation of Nrf2-dependent antioxidant response pathways (Fig. 3c). Surprisingly, our data suggests that only GA and not GB is protective against PQ-induced neurotoxicity. Overall, these data confirm that the neuroprotective effects of GA against PQ toxicity are not solely dependent on the antioxidant activities of flavonoids.

**GardeninA confers neuroprotection by regulating the neuroinflammatory responses in *Drosophila*.** In recent years, the role of dysregulated neuroinflammation in PD pathogenesis has garnered increased attention as an effective way to prevent or delay the onset and progression of NDs[36]. We previously found that PQ exposure activates the innate immune transcription factor, Relish, the *Drosophila* orthologue of mammalian NFκB which in turn leads to reduced survival, increased mobility defects and loss of dopaminergic neurons[6]. Therefore, we next determined the effect of GA and GB on the transcript levels of *relish*. As reported earlier, *relish* transcript levels were induced 2.9-fold in response to PQ treatment (Fig. 4a)[6]. Interestingly, pretreatment with GA reduced the levels of *relish* transcripts to 1.5-fold

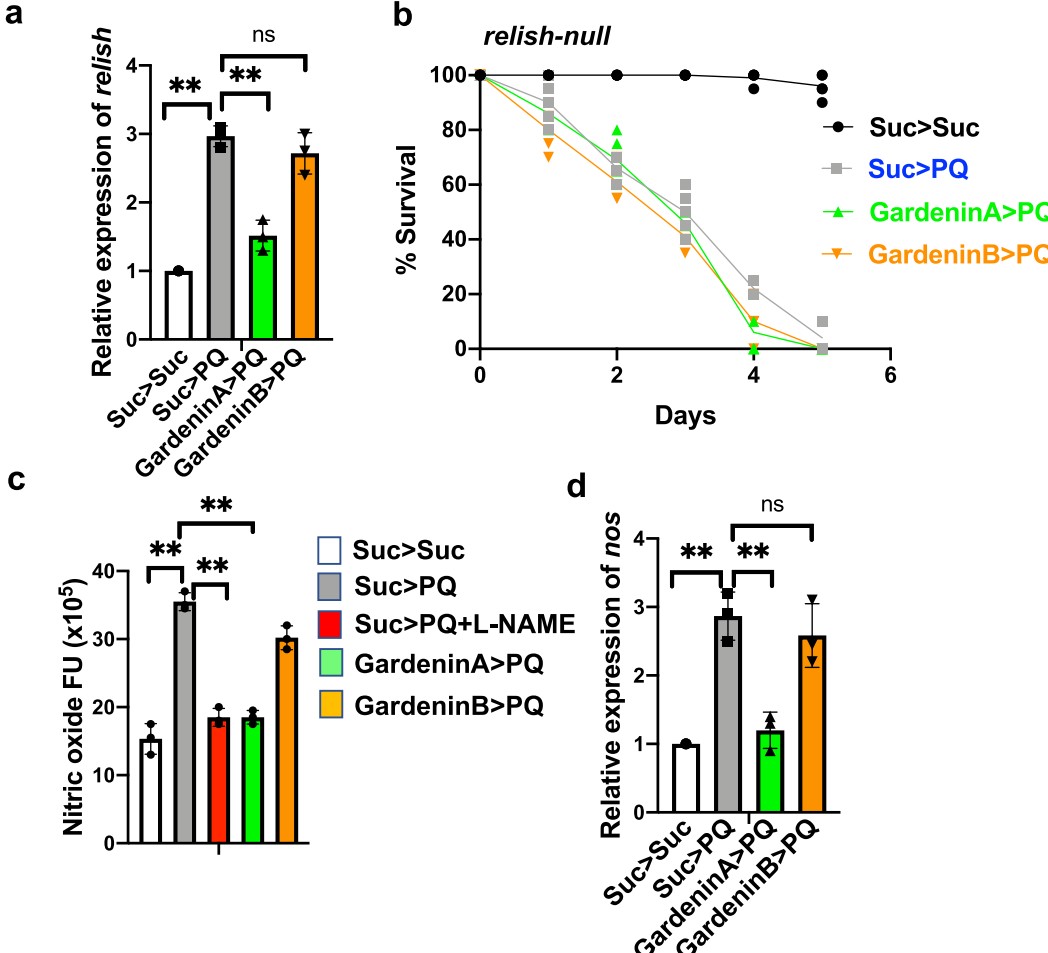

**Fig. 4 GardeninA confers protection against PQ-induced neurotoxicity through modulation of the neuroinflammatory responses in *Drosophila*. a** Effects of GA and GB on the transcript levels of *relish*, the human NFκB orthologue in *Drosophila* using qRT-PCR. RNA was isolated from the heads of adult flies following exposure to the specified feeding conditions. The transcript levels of *relish* were analyzed and plotted after normalization with *rp49* levels as the internal control. Each data point represents mean ± SEM. The mRNA fold changes are normalized to the sucrose-fed (Suc > Suc) flies (assigned a value of 1). \*\*$p < 0.01$ between different feeding conditions based on Mann–Whitney $U$ test. **b** Survival assays were set up using adult *relish-null* mutant male flies following exposure to the indicated feeding conditions and the number of live flies was recorded every 24 h until all of the flies died, and the survival percentages were plotted. Data are representative of ten independent biological replicates with ten male flies per feeding conditions. Data shown represents mean ± SEM. **c** Effects of GA and GB on the levels of nitric oxide using fluorometric assay. Adult male flies were either pre-fed for 4 days with 2.5% sucrose or GA or GB (10 μM) followed by exposure to either sucrose or 5 mM PQ or PQ + L-NAME for 18 h (white bar: Suc>Suc, gray bar: Suc>PQ, red bar: Suc>PQ + L-NAME, green bar: GardeninA>PQ, orange bar: GardeninB>PQ). The fly head homogenates were used to detect the nitrite levels. All experiments were performed in triplicate and represent at least three independent biological repeats. Data shown are expressed as fluorescence arbitrary units (mean ± SEM). \*$p < 0.05$; \*\*$p < 0.01$ based on one-way ANOVA between different feeding conditions. **d** Effects of GA and GB on the transcript levels of nitric oxide synthase (*nos*), the enzyme involved in the synthesis of nitric oxide in *Drosophila* using qRT-PCR. RNA was isolated from the heads of adult flies following exposure to the specified feeding conditions. The transcript levels of *nos* were analyzed and plotted after normalization with *rp49* levels as the internal control. Each data point represents mean ± SEM. The mRNA fold changes are normalized to the sucrose-fed (Suc > Suc) flies (assigned a value of 1). \*\*$p < 0.01$ between different feeding conditions based on Mann–Whitney $U$ test.

but GB failed to suppress the PQ-mediated induction of *relish* transcripts (Fig. 4a). This data suggests that GA-mediated protection against PQ toxicity is dependent on the downregulation of the transcription factor, Relish. To further confirm the involvement of Relish, we performed survival assays in *relish-null* mutant flies that lack the functional Relish protein. We used the same feeding regimen in which the male flies were pre-fed for 4 days with either sucrose as control or 10 µM GA diluted in 2.5% sucrose solution, and then exposed to 5 mM PQ and survival was scored every 24 h and plotted for different feeding groups. As shown in Fig. 4b, pretreatment with GA or GB failed to provide protection against PQ exposure in the absence of Relish, thereby supporting the role of Relish in GA-mediated protection against PQ toxicity.

Previous studies have implicated a role of nitric oxide (NO) in the neuroinflammatory responses in PD pathogenesis[37,38]. In *Drosophila*, NO is produced by hemocytes which are analogous to mammalian macrophages and functions as a critical component of the innate immunity to protect against infection. Interestingly, several studies have shown induction of NO in response to the herbicide, PQ in flies and mammalian PD models[39–41]. Therefore, we investigated the effect of GA and GB pretreatment on NO levels following PQ exposure. Adult flies were pre-fed for 4 days with either sucrose as control or GA or GB 10 µM diluted in 2.5% sucrose solution, and then exposed to 5 mM PQ and NO levels were evaluated in the fly heads using a fluorometric NO assay kit. As shown in Fig. 4c, the nitrite levels were significantly induced by PQ exposure compared to the sucrose-fed control flies ($p < 0.01$). We used NG-nitro-L-arginine methyl ester (L-NAME), a NO synthase (NOS) inhibitor, as a positive control to determine the specificity of the assay. As expected, L-NAME treatment reduced the levels of PQ-induced NO levels. Although, both GA and GB were able to reduce the NO levels, GA was significantly more effective in inhibiting the NO production mediated by PQ.

Next, we determined the effect of GA and GB pretreatment on the transcript levels of NOS, the enzyme responsible for the generation of NO. As shown in Fig. 4d, the levels of NOS mRNA were significantly induced in response to PQ exposure and GA pretreatment reduced the levels of NOS compared to GB, thereby mirroring the results of the biochemical assays for NO generation. Collectively, our data confirms that GA elicits neuroprotection against PQ toxicity through the regulation of inflammatory pathways involving Relish and NO.

**Differential regulation of caspase-dependent cellular death by GardeninA and GardeninB.** Previous studies have indicated the involvement of caspases in inducing the death of DA neurons leading to PD pathogenesis[42,43]. Specifically, the activation of one of the downstream executioner caspases, caspase-3 has been associated with apoptotic cell death resulting in neuronal demise[44]. Moreover, caspase inhibition has been shown to alleviate dopaminergic neuronal death in mammalian PD models. Flavonoids are known to regulate apoptotic pathways in both in vitro and in vivo disease models, including NDs[45,46]. Therefore, we aimed to determine the effects of GA and GB on the activation of caspase-3 responsible for inducing apoptotic cell death. Adult flies were pre-fed for 4 days with either sucrose as control or GA or GB 10 µM diluted in 2.5% sucrose solution, and then exposed to 5 mM PQ followed by detection of caspase-3 activities in the fly heads after 24 h. Consistent with earlier findings, PQ exposure resulted in significant activation of caspase-3 activities when normalized to the sucrose-fed control flies (Fig. 5a). We used a caspase-3 inhibitor, Ac-DEVD-CHO as a positive control to ensure the specificity of the assay. Interestingly, GA pretreatment reduced the levels of PQ-mediated active caspase-3 but GB failed to decrease the levels of caspase-3 activity induced in response to PQ exposure. Furthermore, we determined the role of GA and GB in regulating the expression of an

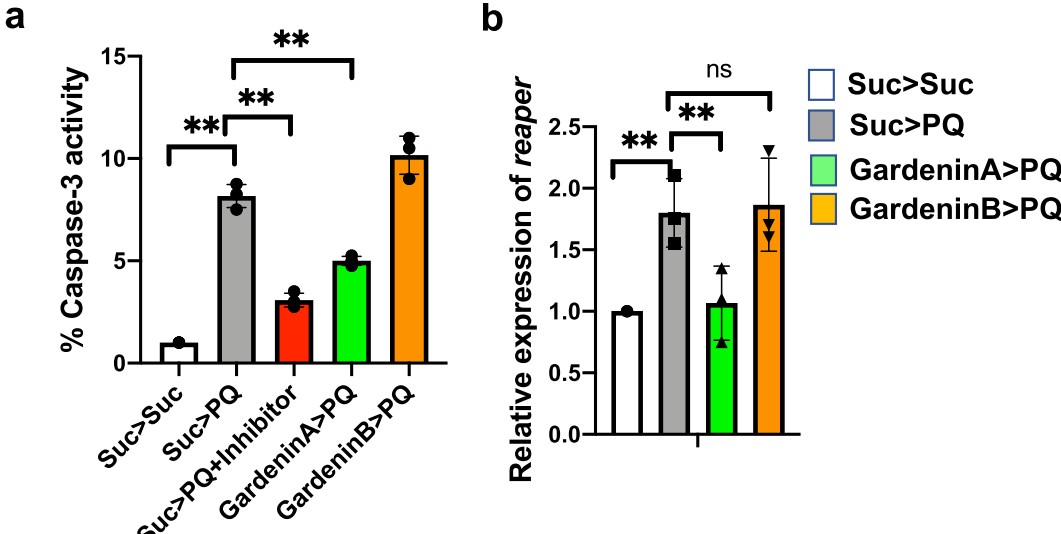

**Fig. 5 Effects of GardeninA and GardeninB on the caspase dependent apoptotic responses in *Drosophila*. a** Caspase-3 activity was measured in response to GA and GB using fluorometric assay. Adult male flies were either pre-fed for 4 days with 2.5% sucrose or GA or GB (10 µM) followed by exposure to either sucrose or 5 mM PQ or PQ + caspase-3 inhibitor (Ac-DEVD-CHO) for 24 h (white bar: Suc>Suc, gray bar: Suc>PQ, red bar: Suc>PQ + inhibitor, green bar: GardeninA>PQ, orange bar: GardeninB>PQ). The fly head homogenates were used to detect the caspase-3 activities. Data are expressed as percentage in relation to the Suc>Suc control and represent averages from at least three independent biological repeats (mean ± SEM). **$p < 0.01$ based on one-way ANOVA between different feeding conditions. **b** Effects of GA and GB on the transcript levels of *reaper*, the upstream proapoptotic gene in *Drosophila* using qRT-PCR. RNA was isolated from the heads of adult flies following exposure to the specified feeding conditions. The transcript levels of *reaper* were analyzed and plotted after normalization with *rp49* levels as the internal control. Each data point represents mean ± SEM. The mRNA fold changes are normalized to the sucrose-fed (Suc > Suc) flies (assigned a value of 1). **$p < 0.01$; ns: not significant between different feeding conditions based on Mann–Whitney $U$ test.

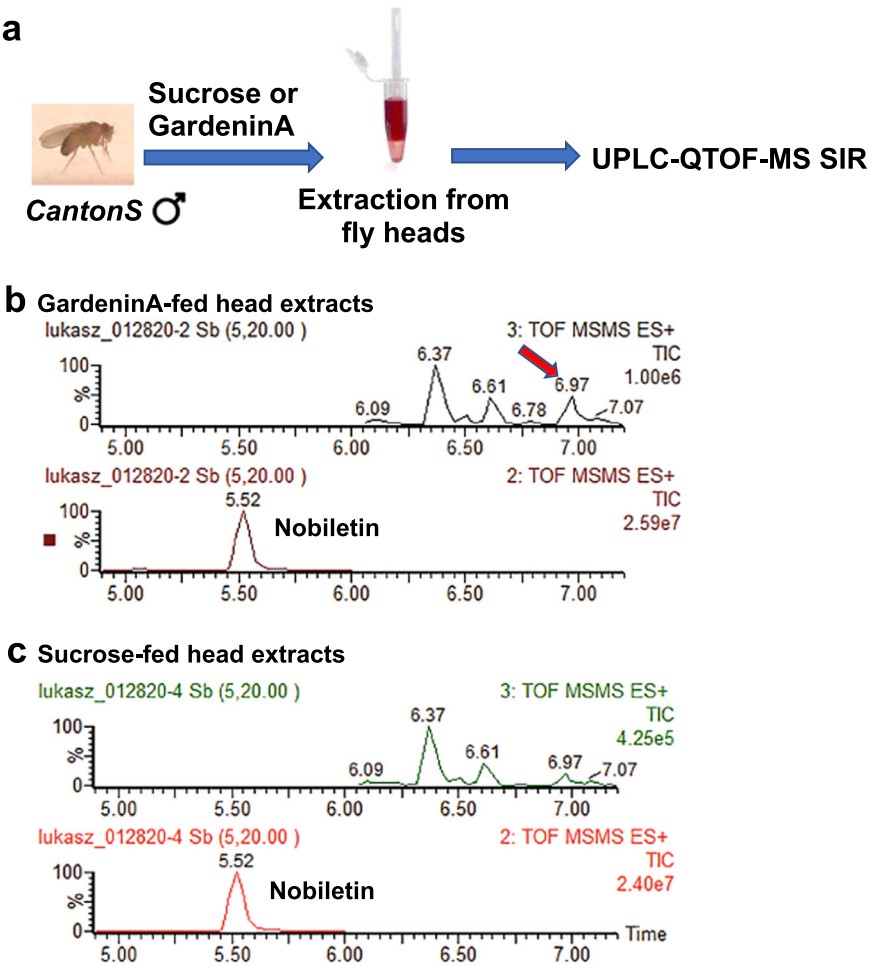

**Fig. 6 Detection of GardeninA in the *Drosophila* heads using UPLC/QTOF/MS SIR. a** Schematic representation of the feeding regimen. Adult male flies were either pre-fed for 4 days with 2.5% sucrose or GardeninA (10 μM) followed by preparation of head homogenates using DMSO. Chromatograms of head homogenates from adult male flies pre-fed for 4 days with either **b** 10 μM GardeninA or **c** 2.5% sucrose followed by extraction and analysis using UPLC/QTOF/MS in positive ionization mode.

upstream proapoptotic gene, *reaper*. In *Drosophila*, reaper can induce apoptosis by inactivating the IAP and thereby rescuing caspases from proteasomal degradation. Using qRT-PCR, we showed that PQ induces the transcript levels of *reaper* and GA pretreatment resulted in the downregulation of the proapoptotic gene, *reaper* (Fig. 5b). Overall, the data confirms that GA suppresses PQ-induced neurodegeneration by inhibiting the activation of proapoptotic factors, including *reaper* and Caspase-3 responsible for dopaminergic neuronal death.

**Quantitation of Gardenin A in the *Drosophila* heads using UPLC/QTOF/MS SIR.** One of the major drawbacks of drug development from natural products is the lack of information regarding the bioavailability of specialized metabolites in both cellular and invertebrate models, including *Drosophila*. We have successfully employed ultra-performance liquid chromatography quadrupole time-of-flight mass spectrometry selected ion recording (UPLC-QTOF-MS SIR) to detect the levels of GA in the fly heads after feeding the flies for 4 days with either sucrose or GA 10 μM as shown in Fig. 6a. The fly heads were collected using liquid nitrogen and homogenized in dimethyl sulfoxide (DMSO) containing specified amounts (1 μM) of a structurally related flavonoid, Nobiletin as the internal standard spiked during the extraction step. The samples were then centrifuged to remove

the debris and the supernatant was collected for analysis. The calibration curves were generated using GA standards in the following concentrations ranging from 1 to 100 nM containing fixed amount of the internal standard Nobiletin (1 μM). The chromatogram profiles in Fig. 6b, c shows the peak of GA and Nobiletin at the retention times of 6.97 min; m/z 419.1342 and 5.52 min; m/z 403.1393, respectively. We employed the peak integration method to calculate the peak area ratios for GA and Nobiletin and detected the average concentration of GA in the fly heads for the GA-fed samples as 7.64 ± 2.1 nM/mg (Fig. 6b). GardeninA is characterized by relatively high lipophilicity (XLogP-AA, 3.2) and strong retention on the chromatographic column, which resulted in traces of this molecule detected in the sucrose-fed samples (Fig. 6c). Based on the standard calibration curve with the peak integration method and linear regression coefficients greater than 0.998, the recovery of the internal standard Nobiletin in the fly head extract was 47 ± 1.45% (*n* = 3). Although characterized by rather low recovery rate, the analytical approach allowed for the detection and quantification of the tested molecule at pharmacologically relevant concentration in the *Drosophila* head samples. The proposed approach still needs further improvement and full validation, but it served the purpose of detecting the presence of GardeninA in the fly heads after oral administration. The analysis of xenobiotics in *Drosophila* is notoriously difficult and there are no available analytical

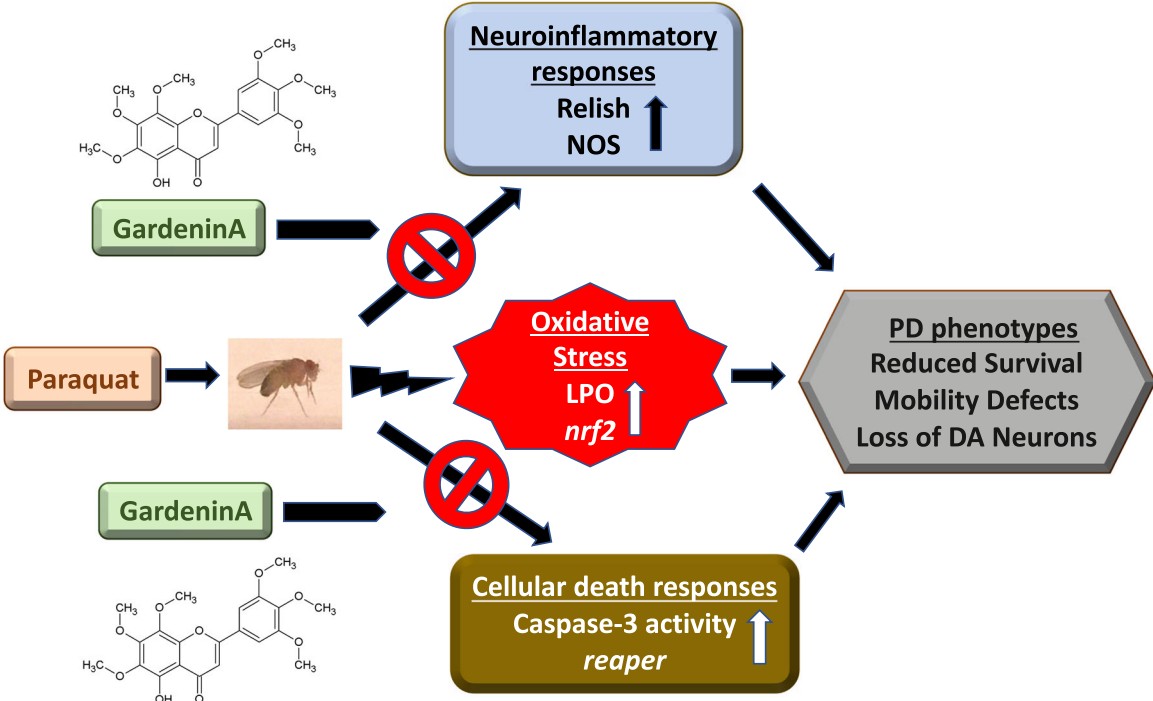

**Fig. 7 Schematic representation of GardeninA-mediated neuroprotection against PQ-induced PD phenotypes in *Drosophila*.** PQ exposure induces oxidative stress and leads to the activation of genes involved in the neuroinflammatory pathways, including Relish and nitric oxide in adult male flies. Furthermore, it activates the proapoptotic responses leading to PD symptoms involving increased mortality, mobility defects and progressive demise of dopaminergic neurons. The flavonoid, GardeninA confers protection against PQ-induced neurotoxicity by inhibiting the neuroinflammatory and the cellular death response pathways, thereby providing protection against PD like symptoms. Interestingly, the antioxidant activity of GardeninA alone is not sufficient to provide protection against PQ-induced neurotoxicity. GardeninA acts on multiple pathways that are commonly associated with PD pathogenesis to reduce the neurotoxic impact of PQ in adult flies. Therefore, Gardenin A can be a potential multifactorial therapeutic for the prevention of PD caused by environmental toxins.

protocols to study the bioavailability of the tested molecules. To the best of our knowledge, this is the first report showing that the oral administration of flavonoids to the flies in lower micromolar concentrations (10 μM) translates to pharmacologically relevant nanomolar dosages in vivo in the fly heads.

## Discussion

In this study, we employed an environmental toxin-induced PD model in *Drosophila* as the screening platform to identify neuroprotective natural products. Current PD therapies are only effective in providing temporary symptomatic relief[10]. There is an urgent need for novel advances in PD therapy that can target the underlying pathways associated with the disease onset and progression. Dietary intake of flavonoids which are naturally occurring plant polyphenols has been shown to alleviate the risk of developing several chronic diseases, including NDs[10,13,47]. We identified the polymethoxyflavonoid, GardeninA (GA), with neuroprotective potential against PQ-induced parkinsonian symptoms along with underlying mechanism of action for improved therapeutic intervention (Fig. 7).

Our data show that pretreatment with the flavonoid, GA, significantly improves survival, mobility defects, and dopaminergic neuron loss against PQ-induced neurotoxicity in *Drosophila*. Another group reported the involvement of GA in promoting neuritogenesis in PC12 cells and neurite alterations have been observed in PD patients[28,48]. On the other hand, the structurally related GardeninB (GB) failed to protect against PQ-induced neurotoxicity, which could potentially be attributed to the lower number of methoxy groups attached to the parent flavone

skeleton. Interestingly, methoxylation of flavones has been shown to enhance their anti-inflammatory activities[49]. PQ is known to induce oxidative stress and dopaminergic neuronal death in both mammalian and *Drosophila* PD models[6,31,50,51]. Oxidative stress has been shown to contribute to the neuronal degeneration in PD pathogenesis, and flavonoids are known to rescue neuronal degeneration[33,52]. However, most attention has been directed toward the antioxidant properties of flavonoids in conferring protection against diverse disease pathology, including NDs[13,34]. Both GA and GB belong to a group of specialized plant metabolites, polyphenols, which previously were considered to confer neuroprotection either by directly scavenging free radicals or by engaging the Nrf2-dependent antioxidant response pathways[13,14]. Previous studies indicate that micromolar concentrations of antioxidants are required to directly scavenge newly formed free radicals in cell culture models, a concentration range far exceeding the physiological levels[35]. Interestingly, our data show that the antioxidant properties of flavonoids are not sufficient to provide protection against environmental toxin-induced neurodegeneration in *Drosophila*. Our data indicate that both GA and GB display antioxidant properties by reducing PQ-mediated lipid peroxidation and inducing the levels of *cncC* and *gstD1* transcripts in the fly heads. A recent study confirmed the antioxidant properties of GA through the suppression of reactive oxygen species in murine EL-4 cells[53]. However, since only GA and not GB elicits neuroprotection against PQ-induced toxicity, thereby confirming that the therapeutic potential is not solely dependent on the ability of GA to reduce PQ-induced oxidative stress. Our data also further confirm that the upregulation of the antioxidant response pathways, such as Nrf2 and its downstream target gene,

*gstD1* are insufficient in providing neuroprotection against PQ-induced neurodegeneration in *Drosophila*.

The emerging evidence suggests that the neuroprotective activity of natural compounds could be attributed to their direct interactions with evolutionary conserved adaptive stress response pathways, as postulated by the neurohormesis hypothesis[13]. This hypothesis implies that most plant-derived specialized metabolites evolved as deterrents and protection at hormetic doses by causing mild cellular damage that leads to the engagement of different stress response pathways[13]. The neurohormesis hypothesis fits well while explaining the neuroprotective potential of noxious compounds at lower concentrations, but it fails to correctly predict the biological activity of polyphenols. Vast majority of polyphenols exhibit low toxicity, even at high concentrations and do not cause cellular damage at hormetic dosages, as suggested by the hypothesis[14].

Sinclair and collaborators proposed that certain phytochemicals, mostly polyphenols may have evolved as chemical cues between autotrophs, herbivores, and omnivores, which became the basis of the xenohormesis hypothesis[14,54]. Stressed plants increase the synthesis of polyphenols which subsequently act as chemical signals allowing plant-feeding animals to sense environmental changes, such as shrinkage of food resources and prepare to face the adversity. Sirtuins are postulated to be majorly responsible for sensing these chemical cues and orchestrating the process of stress response in animals[14]. Polyphenols are synthesized by all plant species and it is not clear which classes of these specialized metabolites work as chemical cues. High doses of polyphenols are required to modulate the postulated cellular targets, often exceeding physiologically relevant concentrations. In addition, certain polyphenols which provided cellular protection did not directly activate sirtuins[55]. Polyphenols, observed to be upregulated under stressful conditions, also failed to activate plant sirtuins[56]. Therefore, the beneficial effects of polyphenol intake may be associated with their interactions with other evolutionary conserved pathways and not necessarily sirtuins.

Our data indicate that GA confers neuroprotection through modulation of the neuroinflammatory pathways involving Relish, a homolog of mammalian NF-κB. We have previously shown that the NF-κB transcription factor, Relish contributes to environmental toxin-induced neurodegeneration in *Drosophila*[6]. NF-κB plays a critical role in regulating the inflammatory responses in both mammalian and *Drosophila* models[57]. Emerging evidence suggests that chronic neuroinflammation plays an essential role in the pathogenesis of diverse NDs, including PD[36,58]. Post-mortem studies of brains of PD patients indicate elevated activation of NF-κB, thereby linking dysregulated inflammatory responses and PD pathogenesis[59]. Similarly, increased NF-κB activation has been associated with age-related neurodegeneration in *Drosophila*[60]. Given the critical association of neuroinflammation with the onset and progression of PD, it is desirable to identify potential drug leads that can target inflammatory pathways involved in the disease pathology. Consistent with our findings, selective inhibition of NF-κB has been shown to be a potential therapeutic target against neurodegeneration in both murine and *Drosophila* PD models[61]. GardeninA appears to modulate oxidative stress and neuroinflammatory responses involving Nrf2 and NF-κB that are known to be associated with PD pathogenesis and are viable targets for therapeutic intervention[62]. Further studies are warranted to elucidate the precise mechanism underlying GA-mediated regulation of crosstalk between Nrf2 and NF-κB against PQ-induced toxicity.

We also observed upregulation of NOS, the enzyme responsible for NO production, and subsequent elevation of NO levels in response to PQ-induced neurodegeneration in *Drosophila*[41]. NO is recognized as a molecular marker for neuroinflammation and

has been shown to be associated with PD pathogenesis in both mammalian and *Drosophila* models[38–40]. Higher levels of inducible NOS (iNOS) were observed in the *substantia nigra* of PD patients and animal models[37,38]. Interestingly, GA pretreatment led to the downregulation of both NOS and NO levels induced by exposure to PQ, thereby providing protection against NO-mediated inflammatory responses. Consistent with our data, another structurally related flavonoid from citrus fruit peels, Tangeretin, has been shown to inhibit lipopolysaccharide-induced production of NO to confer neuroprotection in several ND models[63]. Surprisingly, plant cells also experience oxidative burst and elevation of NO levels upon PQ exposure[64,65]. Increasing levels of reactive oxygen species and NO result in PQ-induced cell death[66]. Furthermore, NO has been previously shown to induce genes required for the synthesis of polyphenols that are known to confer protection against PQ[67]. However, it still remains unclear if the innate immune systems in plants and animals evolved from a common ancestor or if the observed functional overlap stems from convergent evolution[68]. Little attention has been given to understanding the role of stress-induced polyphenols in the regulation of innate immunity systems and programmed cell death in plants and animals. We speculate that certain plant polyphenols may be involved in the regulation of molecular pathways leading to cell death during innate immune responses in plants and animals. Stafford previously argued that certain polyphenols may have evolved as internal physiological regulators or chemical messengers[69]. Further studies are needed to better understand the role of these polyphenols in the regulation of plant and animal innate immunity. It is of crucial importance to study these interactions in the context of a live organism at physiologically relevant concentrations.

Our data also demonstrate that GA pretreatment can reduce the levels of PQ-induced proapoptotic factors, including caspase-3 and *reaper*. Dysregulated caspase activation underlies several chronic disease conditions, including NDs[43]. Drosophila Reaper is known to bind to the inhibitor of apoptosis proteins (IAPs) to rescue caspases from proteasomal degradation and promote apoptosis[70]. Caspase activation appears to be involved in the progressive demise of the dopaminergic neurons resulting in PD pathogenesis. Previous studies have shown that flavonoids can function as specific activators or inhibitors of caspases[45,46]. Inhibitors against different caspases have been shown to rescue neuronal deaths in several experimental PD models[42,71]. Interestingly, caspase-3 dependent dopaminergic cell death has been observed in *Drosophila* models of PD[44,72]. Moreover, the involvement of caspase-8 has been reported in both human PD patients and mammalian mouse PD models[73]. Recently, another flavonoid kaempferol has been shown to inhibit caspase-3 activities in a genetic model of PD in flies[74]. However, the ability of GardeninA to regulate multiple pathways involving oxidative stress, neuroinflammatory and cellular death responses that are intricately linked to PD pathogenesis makes it a promising therapeutic target. Interestingly, our data also show that GB can induce caspase-3 activities in *Drosophila* heads. In support of our findings, GB has been shown to induce caspase-dependent apoptosis in human leukemia cell lines[75]. Future studies will be directed toward exploring the structural features of flavonoids that are critical for modulation of caspase activities in PD models.

Most studies involving natural products derived therapeutics translates to poor clinical outcome due to insufficient data regarding the bioavailability. This is the first report to reveal the presence of GA in the *Drosophila* heads in pharmacologically relevant nanomolar doses following food uptake containing 10 μM GA. Although we did not test the bioavailability of GB in the *Drosophila* heads since it was not protective in our PD model,

we speculate its presence in the fly heads due to its ability to induce the antioxidant and caspase-3 activities in response to PQ. We want to point out one of the limitations of the study in that we did not measure any other compounds that form as a result of the metabolism of GA in the fly gut, which might also influence the GA bioavailability in the brain. A recent study analyzed the in vivo metabolic profile of GA in rat plasma using high-performance liquid chromatography coupled with linear ion trap–Orbitrap mass spectrometer[76]. Further studies are required to detect the presence of other products resulting from in vivo metabolic biotransformation of flavonoids for improved therapeutic efficacy. Another problem in translational studies concerning the use of polyphenols is the fact they often are excluded from serious considerations as drug leads. This stems from their common classification as so called PAINS (pan-assay interference compounds) attributed to their nonspecific interactions with multiple protein targets[77]. The majority of these interactions have been reported based on screening large libraries of individual compounds using high-throughput cell-based assays, often at super-physiological concentrations, thereby underscoring the importance of testing these molecules in the context of a live organism at physiologically relevant concentrations. In addition, it should not be excluded that certain polyphenols may have evolved to interact with multiple protein targets[10,14].

Although, the exact etiology of sporadic PD pathogenesis remains elusive, it is thought to be multifactorial involving genetic and environmental triggers associated with disease onset. Earlier studies on drug discovery mainly focused on single target to avoid unnecessary side effects and toxicity[78,79]. Due to the multifaceted nature of NDs, it is necessary to pursue multiple target approach to achieve improved therapeutic outcomes. Our data identified GA as a potential multi-target neuroprotective agent against environmental toxin-induced PD symptoms in *Drosophila* through its diverse pharmacological effects, including antioxidant, anti-inflammatory, and antiapoptotic properties (Fig. 7). Future studies are needed in genetic and mammalian PD models to validate the neuroprotective effects of GA for positive outcomes in clinical trials.

## Methods

**Drosophila culture and stocks**. The fly stocks were raised on standard medium (Bloomington Stock Center recipe) containing cornmeal, corn sirup, yeast, and agar at 25 °C under a 12 h of light and 12 h of darkness cycle. The wild-type strain *Canton S* was used as the control genotype for all experiments. Male flies aged 3–5 days old post-eclosion were used for all assays. The following stocks were purchased from the Bloomington Drosophila Stock Center at Indiana University: *CantonS, w^{1118}; Rel^{E20}*.

**Chemicals**. The following chemicals were purchased from Sigma-Aldrich: Sucrose, PQ (methyl viologen dichloride hydrate), GardeninA, GardeninB, NOS inhibitor L-NAME. ProLong Gold Antifade Mountant with DAPI and TRIzol reagents were obtained from Thermo Fisher Scientific.

**Drosophila feeding regimen**. Ten males of specified genotype, aged 3–5 days post eclosion, were maintained in vials and fed daily on filter paper saturated with specified concentrations of PQ or flavonoids (GardeninA or GardeninB) in 2.5% sucrose or with 2.5% sucrose only. Mortality was monitored daily until all flies were dead. Survival assays were performed with at least ten independent biological replicates of ten males per vial for different feeding conditions.

**Mobility assay**. Negative geotaxis assay was used to determine the mobility defects in adult male flies from each treatment group. Ten flies per feeding condition were placed in an empty plastic vial and gently tapped to the bottom. The percentage of flies that crossed a line 5 cm from the bottom of the vial in 20 s were recorded to compare between different feeding conditions. Ten independent biological replicates were assayed three times at 5-min intervals and the percentages averaged. Statistical significance between different genotypes was calculated using one-way analysis of variance (ANOVA) for $p < 0.05$.

**Immunofluorescence analysis**. Intact brains were dissected from adult male flies exposed to different feeding conditions as specified and processed for confocal imaging as described earlier[6]. Briefly, brains were dissected in 1× phosphate buffered saline with Tween20 (PBST), fixed in 4% paraformaldehyde for 20 min in a 24-well plate, and washed three times using 1× PBST. Brains were mounted on slides with coverslips using ProLong Gold Antifade Mountant with DAPI (Thermo Fisher Scientific). Whole mounts of dissected brains were imaged, and GFP-positive dopaminergic neurons were detected using a Nikon C2 DUVb confocal laser scanning microscope. The GFP-positive cells of specific DA clusters were counted based on confocal Z-stacks of whole mount brains.

**Quantitative real time RT-PCR**. Total RNA was extracted from the heads of 25–30 adult male flies using TRIzol Reagent, and cDNA was prepared from 0.5 to 1 µg total RNA using the High Capacity cDNA Reverse Transcription kit (Applied Biosystems). The resulting cDNA samples were amplified using the iQ SYBR Green Supermix (Biorad) in a StepOnePlus Real time PCR System according to the manufacturer's protocols. The relative levels of specified transcripts were calculated using the ΔΔCt method, and the results were normalized based on the expression of *ribosomal protein L32* (*RpL32/RP49*) as the endogenous control within the same experimental setting. At least three independent biological replicates were performed in response to specified feeding conditions for data analysis.

**NO assay**. The levels of NO were measured using a fluorometric assay kit according to the manufacturer's protocol (Cayman Chemical). Briefly, twenty fly heads were collected using liquid nitrogen from specified feeding conditions and homogenized in the assay buffer supplied with the kit. The resulting homogenates were centrifuged, and the supernatant were used for analysis using a 96-well white plate. The nitrite levels were determined with the addition of 2,3-diamino-naphthalene (DAN) and sodium hydroxide which results in the formation of a fluorescent product, 1(H)-naphthotriazole. The plates were then read in a fluorometer SpectroMax I3X (Molecular Devices) using an excitation wavelength of 360 nm and an emission wavelength of 430 nm.

**Lipid peroxidation assay**. The levels of oxidative stress were determined using a fluorometric assay kit according to the manufacturer's protocol (Cayman Chemical). Briefly, twenty fly heads were collected using liquid nitrogen from specified feeding conditions and homogenized in ice-cold RIPA buffer followed by centrifugation at $1600 \times g$ for 10 min at 4 °C. The resulting head homogenates were used to detect the levels of MDA, which is a naturally occurring product of lipid peroxidation, using a fluorometer SpectroMax I3X (Molecular Devices) at an excitation wavelength of 530 nm and an emission wavelength of 550 nm.

**Caspase-3 activity assay**. The assay was performed according to the manufacturer's protocol using the EnzChek Caspase-3 assay kit (Thermo Fisher Scientific). Briefly, homogenates were prepared from ten fly heads per feeding conditions using ice-cold cell lysis buffer supplied with the kit. The reaction was initiated using the Z-DEVD-AMC substrate and incubated at room temperature for 30 min. The inhibitor Ac-DEVD-CHO was added in specified wells to confirm the specificity of the assay. The plates were then read in a fluorometer SpectroMax I3X (Molecular Devices) using an excitation wavelength of 342 nm and an emission wavelength of 441 nm.

**UPLC-QTOF-MS SIR analysis of *Drosophila* head extracts**. Adult male flies (250 per feeding conditions; three independent biological replicates) were fed either sucrose as a control or 10 µM GardeninA for 4 days, followed by preparation of extracts from the fly heads using DMSO. Internal standard Nobiletin (1 µM) was added during the homogenization step. The undiluted head homogenates in DMSO were directly used for analysis. UPLC-MS SIR analysis was performed using a Waters Xevo G2-XS QTOF mass spectrometer coupled with an ACUITY I-Class UPLC system. An ACUITY UPLC BEH C18 column (2.1 × 50 mm, 1.7 µm) was utilized with an ACQUITY UPLC BEH C18 Van Guard Pre-column (2.1 × 50 mm, 1.7 µm). The column was eluted using the following gradient solutions at a flow rate of 0.3 mL/min with mobile phase A (water with 0.1% formic acid) and B (acetonitrile with 0.1% formic acid): 0.0 min 80% A 20% B, 5.0 min 60% A 40% B, 10.0–15.0 min 0% A 100% B, 16.0–19 min 80% A 20% B. The injection volume for each sample was 5 µL. SIR was performed in resolution positive ion mode on [M + H]$^+$ ions of Nobiletin m/z 403.1393 during 0.0-6.0 min and GardeninA m/z 419.1342 during 6.0–19.0 min of the chromatogram. The ESI source conditions include: Capillary voltage 0.35 kV, Sampling cone 40 V, Source offset 80, Source temperature 100 °C, Desolvation 350 °C, Cone gas 0 L/h, Desolvation gas 400 L/h, lock spray capillary voltage 0.08 kV, lock mass: 556.2771 in positive ionization mode (leucine enkephalin, peak resolution 28,000). To quantify GardeninA concentrations in the fly head extracts, an internal calibration curve was constructed by plotting the integrated chromatogram peak area ratio (no peak smoothing) of GardeninA calibration standards (1, 10, 25, 50, and 100 nM) and internal standard Nobiletin (1 µM) versus their concentration ratio. To calculate % recovery, an external calibration curve of Nobiletin was constructed by plotting the integrated chromatogram peak areas (no peak smoothing) versus Nobiletin concentrations (50, 100, 250, 500, 1000 nM). The final concentrations of GardeninA in the head

homogenates were calculated based on the total amounts per mg ±SD of fly heads for different feeding conditions.

**Statistics and reproducibility**. Data were analyzed using the GraphPad Prism 8 software (GraphPad Software, Inc., La Jolla, CA). Statistical significances of gene expression between different groups were determined using a non-parametric Mann–Whitney $U$ test. For mobility and immunofluorescence assays, one-way analysis of variance (ANOVA) was used to compare differences between specified feeding groups. Results are expressed as mean ± SEM, and $p < 0.05$ was considered statistically significant. Details of the specific statistical analyses and biological replicates are indicated in the figure legends.

**Reporting summary**. Further information on research design is available in the Nature Research Reporting Summary linked to this article.

## Data availability

All the data generated or analyzed during this study are included in this published article or available from the corresponding author upon request.

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

## Acknowledgements
This work was supported by start-up funds from the University of Alabama, American Society of Pharmacognosy Research Starter Grant and Alabama Life Research Institute grant to L.C.

## Author contributions
U.M. and L.C. designed the research; U.M. and T.H. performed the experimentations and U.M. analyzed the data; Q.L. performed the QTOF analysis; and U.M. and L.C. wrote the paper. All authors reviewed the paper.

## Competing interests
The authors declare no competing interests.
