## [Peer Review File · Communications Biology]

Reviewers' comments:

Reviewer #1 (Remarks to the Author):

please see attached file

** Copied in by editor**

Report on "GardeninA confers neuroprotection against environmental toxin in a Drosophila model of Parkinson's disease"

This paper is generally well written and clear. It has a specific aim and objectives that hang together. Two natural products GardeninA (GA) and GardeninB (GB) were tested in a fly model of PD. GA, but not GB, has a measurable rescue across a range of parameters.

As the authors themselves note, the impact of a range of natural products on this model suffers from a range of poor controls, pleiotropic effects, and lack of bioavailability. To their credit, the authors attempt to control their experiments efficiently, in part by comparing two related compounds (GA, GB) and in part by examining the uptake of GA into the head. It would be interesting to see how much GB is taken up, in case the difference between GA and GB is due to bioavailability.

The Discussion could include comparison with other natural products, with some critical evaluation of the differences.

Minor points:

Cell counts: in the literature, the number of cells counted in control preparations in the PPL ranges from 9 to 22; PPM from 6-17. Some comment on the authors' counts in this context might be appreciated.

P6: "We have only used adult male flies due to their higher sensitivity to PQ toxicity leading..." -you could also say that females eat differently depending on whether they have been mated.

Figures:

It would make life easier if Figs 1 and 2 were combined, so that the differences in structure and response to PQ between GA and Gb are clear.

No need to underline text – just use bold

In Fig. 5b, the survival curve should be plotted in the same way as in Fig.1, and it should be explicit in the figure (not just the legend) that these are not wild-type flies.

References: there seem to be a lot, and I would have thought for a research paper, the argument could be slimmer. A large number of references would be more suitable for a review.

Reviewer #2 (Remarks to the Author):

The manuscript of Maitra et al is principally highly interesting as it provides a novel compound that interferes with the experimental development of Parkinson-like phenotypes in the fly. GardeninA has these beneficial effects and its characterization is comprehensive.

nevertheless, some questions remain for the manuscript.

1) GardeninA is doing almost everything (Nrf2, antioxidant, Relish, NO). All these pathways and systems are usually not interdependent but independent from each other, how should this work?

2) Nrf2 and relish are transcription factors and their activation is usually more or less independent from the expression level. For the Nrf2 system, the balance of cnc and keap1 is decisive. Moreover, a number of canonical pathway genes are known. Alternatively, reporter lines (GSTD2 or 5) are available that monitor Nrf2 activation. I cannot see any effect of GardeninA on the cnc level (Fig. 4b). Similar problems are relevant for relish - here nuclear translocation monitors its activation.

3) Although GardeninA is so active, why they could not show any effect on lifespan?

4) The experimental design might simply reduce the amount of PQ reaching the brain as some of the PQ might be detoxified in the gut.

5) An important confounder in feeding studies is the effect on feeding rates. Here, GardeninA might reduce feeding rates and therewith ingestion of PQ -- this needs to be checked.

Reviewer #3 (Remarks to the Author):

Paper # COMMSBIO-20-1731-T

The goal of the research article by Urmila et al. is to analyse the neuroprotective potential of the polymethoxyflavonoid, GardeninA (GA), in paraquat (PQ) induced PD model in *Drosophila*. The authors employed array of assays and techniques to monitor behavioural, molecular and biochemical changes. In addition the authors went further to test bioavailability of the GA in the *Drosophila* brains. Further they used immunohistochemistry to monitor the number of DA neurons during PQ induced toxicity and the neuroprotection conferred by GA. The current major goal in PD field is to search for alternative effective therapy and this study is solely aligned with this goal. Overall the work is highly relevant and the authors did utilized different approaches to justify their findings. The work is well written, however I have few minor comments.

Specific comments

1. On the introduction section, since the GA is the focus of this study, I would recommend to introduce it earlier on rather than waiting for the reader to know what is GA only in the discussion part. This can be stated in the 4th paragraph of the introduction where authors are describing plant derived molecules and their use to manage neurodegenerative diseases. One would wish to know which plant GA is isolated from or if it synthetic one. Also if there are any studies on GA that are related to neurodegeneration or neuroprotection would not harm mentioning the key ones in the introductory section.

2. On the methodology section; while reading the manuscript especially in the result section, I realized that all the different techniques or assays used, the feeding or substance administration to flies was similar or the same. For example the concentration of PQ, the time of exposure, concentration of GA, concentration of sucrose etc. I thought instead of repeating the explanation in each technique or assay used, such information can be once captured in details in the methodology

section in the subsection of *Drosophila* feeding regimen, then in the result section to remain with very specific ones.

-Authors did not state anywhere in the manuscript why or what was the criteria of choosing four (4) days for GA pre-feeding prior to PQ exposure, why not 2 or 5. If there are reasons, would be good to state them so that it can be clear to the reader, as it is now one remains with a question why four?

- Authors should check if it is necessary to separate fig. 1c and fig.1e. Both figures are showing negative geotaxis assay and reading on the figure legend of fig 1, authors stated the time points in which the assays were conducted in which both were taken at same time points (24 h and 48 h). I would suggest to combine the graphs, this will also allows a reader to see the effect of L-DOPA in comparison to GA during PQ exposure in one single graph!

3. On the result section on page 7 under the subsection with a title - effect of structurally related GardeninB.....line 4 from top - a word files should be removed. It seems to be typing error.

Report on "GardeninA confers neuroprotection against environmental toxin in a Drosophila model of Parkinson's disease"

This paper is generally well written and clear. It has a specific aim and objectives that hang together. Two natural products GardeninA (GA) and GardeninB (GB) were tested in a fly model of PD. GA, but not GB, has a measurable rescue across a range of parameters.

As the authors themselves note, the impact of a range of natural products on this model suffers from a range of poor controls, pleiotropic effects, and lack of bioavailability. To their credit, the authors attempt to control their experiments efficiently, in part by comparing two related compounds (GA, GB) and in part by examining the uptake of GA into the head. It would be interesting to see how much GB is taken up, in case the difference between GA and GB is due to bioavailability.

The Discussion could include comparison with other natural products, with some critical evaluation of the differences.

Minor points:

Cell counts: in the literature, the number of cells counted in control preparations in the PPL ranges from 9 to 22; PPM from 6-17. Some comment on the authors' counts in this context might be appreciated.

P6: "We have only used adult male flies due to their higher sensitivity to PQ toxicity leading..." -you could also say that females eat differently depending on whether they have been mated.

Figures:

It would make life easier if Figs 1 and 2 were combined, so that the differences in structure and response to PQ between GA and Gb are clear.

No need to underline text - just use bold

In Fig. 5b, the survival curve should be plotted in the same way as in Fig.1, and it should be explicit in the figure (not just the legend) that these are not wild-type flies.

References: there seem to be a lot, and I would have thought for a research paper, the argument could be slimmer. A large number of references would be more suitable for a review.

The goal of the research article by Urmila et al. is to analyse the neuroprotective potential of the polymethoxyflavonoid, GardeninA (GA), in paraquat (PQ) induced PD model in *Drosophila*. The authors employed array of assays and techniques to monitor behavioural, molecular and biochemical changes. In addition the authors went further to test bioavailability of the GA in the *Drosophila* brains. Further they used immunohistochemistry to monitor the number of DA neurons during PQ induced toxicity and the neuroprotection conferred by GA. The current major goal in PD field is to search for alternative effective therapy and this study is solely aligned with this goal. Overall the work is highly relevant and the authors did utilized different approaches to justify their findings. The work is well written, however I have few minor comments.

Specific comments

1. On the introduction section, since the GA is the focus of this study, I would recommend to introduce it earlier on rather than waiting for the reader to know what is GA only in the discussion part. This can be stated in the 4th paragraph of the introduction where authors are describing plant derived molecules and their use to manage neurodegenerative diseases. One would wish to know which plant GA is isolated from or if it synthetic one. Also if there are any studies on GA that are related to neurodegeneration or neuroprotection would not harm mentioning the key ones in the introductory section.
2. On the methodology section; while reading the manuscript especially in the result section, I realized that all the different techniques or assays used, the feeding or substance administration to flies was similar or the same. For example the concentration of PQ, the time of exposure, concentration of GA, concentration of sucrose etc. I thought instead of repeating the explanation in each technique or assay used, such information can be once captured in details in the methodology section in the subsection of *Drosophila* feeding regimen, then in the result section to remain with very specific ones.
 - Authors did not state anywhere in the manuscript why or what was the criteria of choosing four (4) days for GA pre-feeding prior to PQ exposure, why not 2 or 5. If there are reasons, would be good to state them so that it can be clear to the reader, as it is now one remains with a question why four?
 - Authors should check if it is necessary to separate fig. 1c and fig.1e. Both figures are showing negative geotaxis assay and reading on the figure legend of fig 1, authors stated the time points in which the assays were conducted in which both were taken at same time points (24 h and 48 h). I I would suggest to combine the graphs, this will also allows a reader to see the effect of L-DOPA in comparison to GA during PQ exposure in one single graph!
3. On the result section on page 7 under the subsection with a title - **effect of structurally related GardeninB**.....line 4 from top - a word files should be removed. It seems to be typing error.

To
Dr. Luke Grinham,
Associate Editor, *Communications Biology*.

October 9, 2020

Dear Dr. Grinham,

We are thankful for your invitation to revise and resubmit our manuscript titled " GardeninA confers neuroprotection against environmental toxin in a Drosophila model of Parkinson's disease."

Please find below our point-by-point response to the reviewer's comments.

Reviewer#1 (Remarks to the Author):

1. This paper is generally well written and clear. It has a specific aim and objectives that hang together. Two natural products GardeninA (GA) and GardeninB (GB) were tested in a fly model of PD. GA, but not GB, has a measurable rescue across a range of parameters. As the authors themselves note, the impact of a range of natural products on this model suffers from a range of poor controls, pleiotropic effects, and lack of bioavailability. To their credit, the authors attempt to control their experiments efficiently, in part by comparing two related compounds (GA, GB) and in part by examining the uptake of GA into the head. It would be interesting to see how much GB is taken up, in case the difference between GA and GB is due to bioavailability.

1. Author's comments: We thank the reviewer for the positive comments on our manuscript. As mentioned in the Discussion section of the revised manuscript (page 21), we do not think that the difference in activities between GA and GB is due to their bioavailability. We would also like to point out that GB was active in the fly heads due to its ability to induce the antioxidant and caspase-3 activities in response to PQ, which probably suggests its bioavailability in the fly heads. We decided not to pursue the technically challenging experiments to test the bioavailability of GB in the fly heads since it was not protective in our PD model and the results are unlikely to add any meaningful data to the manuscript.

2. The Discussion could include comparison with other natural products, with some critical evaluation of the differences.

2.Author's comments: We have added a few examples of other flavonoids (pages 19 and 20) as compared to GardeninA that appear to regulate one or more molecular pathways associated with PD pathogenesis. However, the ability of GardeninA to regulate multiple pathways that are intricately linked to PD makes it a promising therapeutic target.

3.Minor points:

Cell counts: in the literature, the number of cells counted in control preparations in the PPL ranges from 9 to 22; PPM from 6-17. Some comment on the authors' counts in this context might be appreciated.

3.Author's comments: The difference in the numbers of dopaminergic neuron clusters reported in the literature are due to different modes of detection techniques using either anti-GFP or anti-TH (tyrosine hydroxylase) antibodies. As mentioned in our manuscript, we monitored the DA clusters using GFP expression driven by the tyrosine hydroxylase (TH)-promoter in *TH-Gal4; UAS-GFP* transgenic adults. Moreover, in case of the PPL clusters, we are looking at the PPL1 cluster only that appears to be sensitive to PQ treatment and not PPL2.

Our results are consistent with other reported studies (Journal of Neuroscience, 27 (10) 2457-2467; DOI: <https://doi.org/10.1523/JNEUROSCI.4239-06.2007>; Front Neurosci. 2010; 4: 205. doi: 10.3389/fnins.2010.00205; *Sci Rep* **9**, 12714 (2019). <https://doi.org/10.1038/s41598-019-48977-6>).

4.P6: "We have only used adult male flies due to their higher sensitivity to PQ toxicity leading..." -you could also say that females eat differently depending on whether they have been mated.

4.Author's comments: We agree with the reviewer and have added that comment to the revised manuscript on Page 7 (highlighted text).

5.Figures: It would make life easier if Figs 1 and 2 were combined, so that the differences in structure and response to PQ between GA and Gb are clear. No need to underline text – just use bold.

5. Author's comments: As suggested by the reviewer, we have combined Figs 1 and 2 in the revised Results section and removed the underline text from GardeninB. In addition, we have transferred the data on L-Dopa to Supplemental Fig 1 as this experiment was performed only to highlight the suitability of our PD fly model as a screening platform since it mimics the therapeutic effects observed in human PD patients.

Fig. 1

6.In Fig. 5b, the survival curve should be plotted in the same way as in Fig.1, and it should be explicit in the figure (not just the legend) that these are not wild-type flies.

6. Author's comments: We agree with the reviewer and have mentioned that these are *relish-null mutant* flies in the figure 4b. However, the justification for showing the entire survival curve is to point out that there is no protection at any given day during the entire duration of the treatments with either GA or GB against PQ-induced toxicity.

7.References: there seem to be a lot, and I would have thought for a research paper, the argument could be slimmer. A large number of references would be more suitable for a review.

7. Author's comments: We have tried our best to limit the number of references. We also included a few examples of other flavonoids in the Discussion as suggested by the reviewer (Ref#2 comments), which added to the current reference list.

Reviewer #2 (Remarks to the Author):

The manuscript of Maitra et al is principally highly interesting as it provides a novel compound that interferes with the experimental development of Parkinson-like phenotypes in the fly. GardeninA has these beneficial effects and its characterization is comprehensive. nevertheless, some questions remain for the manuscript.

1. GardeninA is doing almost everything (Nrf2, antioxidant, Relish, NO). All these pathways and systems are usually not interdependent but independent from each other, how should this work?

1. Author's comments: We thank the reviewer for the constructive and positive comments on our manuscript.

Contrary to the reviewer's comments, the Nrf2 and NFkB/Relish are interconnected as per current literature cited below.

(J Clin Cell Immunol 2017, 8:1 DOI: 10.4172/2155-9899.1000489; Biochem Soc Trans (2015) 43 (4): 621–626. <https://doi.org/10.1042/BST20150014>; ASN Neuro. 2020 Jan-Dec; 12: 1759091419899782 doi: 10.1177/1759091419899782).

Both oxidative stress and neuroinflammatory responses are associated with PD pathogenesis and are good targets for therapeutic intervention (Clin Neurosci Res. 2006 Dec 6; 6(5): 261–281. doi: 10.1016/j.cnr.2006.09.006).

NFkB plays an essential role in the neuroinflammatory responses through the regulation of pro-inflammatory cytokines and nitric oxide production in mammalian models. Therefore, the ability of GardeninA to regulate multiple pathways involving oxidative stress, neuroinflammatory and cellular death responses that are intricately linked to PD pathogenesis makes it a promising therapeutic target.

2. Nrf2 and relish are transcription factors and their activation is usually more or less independent from the expression level. For the Nrf2 system, the balance of cnc and keap1 is decisive. Moreover, a number of canonical pathway genes are known. Alternatively, reporter lines (GSTD2 or5) are available that monitor Nrf2 activation. I cannot see any effect of GardeninA on the cnc level (Fig. 4b). Similar problems are relevant for relish - here nuclear translocation monitors its activation.

2. Author's comments: We thank the reviewer for the comments. As suggested by the reviewer, we have performed additional experiments to determine the gene expression level of a stress response gene, glutathione-S-transferase (*gstD1*) downstream of Nrf2 (Fig. 3c). Our data show that GA and GB further increases the transcript levels of both *cncC* and *gstD1* together with PQ due to the antioxidant response against oxidative stress. We followed it up with the lipid peroxidation assays to show the antioxidant activities of GA and GB against PQ-induced oxidative stress, which strongly suggests activation of the antioxidant response pathways involving Nrf2.

Based on our earlier publications, we have observed increase in both mRNA and protein levels of NFkB/Relish in response to PQ exposure (Maitra, U., Scaglione, M. N., Chtarbanova, S. & O'Donnell, J. M. (2019) Innate immune responses to paraquat exposure in a Drosophila model of Parkinson's disease. Sci Rep 9, 12714, doi:10.1038/s41598-019-48977-6). In this manuscript, we

show that only GA and not GB can reduce the levels of *relish* transcripts in response to PQ treatment. Moreover, pre-treatment with GA or GB failed to provide protection against PQ exposure in *relish-null* mutant flies that lacks a functional Relish protein, thereby supporting the role of Relish in GA-mediated protection against PQ toxicity. Our next step would be to evaluate the detailed mechanistic effects of these compounds on the specific neuroinflammatory pathways elicited by specific immune cells (hemocytes, glial cells, etc.) that are associated with PD pathogenesis, which is beyond the scope of this manuscript.

3. Although GardeninA is so active, why they could not show any effect on lifespan?

3. Author's comments: In this manuscript, we have only focused on the ability of GA to improve reduced survival induced by the exposure to the environmental toxin, PQ. We are currently interested to explore the role of GA to enhance lifespan and effects on anti-aging pathways related to neurodegenerative disorders, which is beyond the scope of this manuscript.

4. The experimental design might simply reduce the amount of PQ reaching the brain as some of the PQ might be detoxified in the gut.

4. Author's comments: We have used the dosage of paraquat at 5 mM to induce PD phenotypes in a reproducible manner based on rigorous screening as reported in earlier publications (Maitra, U., Scaglione, M. N., Chtarbanova, S. & O'Donnell, J. M. (2019) Innate immune responses to paraquat exposure in a *Drosophila* model of Parkinson's disease. *Sci Rep* 9, 12714, doi:10.1038/s41598-019-48977-6; Effects of Dual Exposure to the Herbicides Atrazine and Paraquat on Adult Climbing Ability and Longevity in *Drosophila melanogaster*. *Insects* 2019 Nov 10;10(11):398. doi:10.3390/insects10110398.).

5. An important confounder in feeding studies is the effect on feeding rates. Here, GardeninA might reduce feeding rates and therewith ingestion of PQ -- this needs to be checked.

5. Author's comments: We thank the reviewer for this comment and agree with the reviewer. We have performed additional experiments and added data from feeding assays to show that the feeding rates are comparable as reported in the supplemental figure (FigS2).

Reviewer #3 (Remarks to the Author):

Paper # COMMSBIO-20-1731-T

The goal of the research article by Urmila et al. is to analyse the neuroprotective potential of the polymethoxyflavonoid, GardeninA (GA), in paraquat (PQ) induced PD model in Drosophila. The authors employed array of assays and techniques to monitor behavioural, molecular and biochemical changes. In addition the authors went further to test bioavailability of the GA in the Drosophila brains. Further they used immunohistochemistry to monitor the number of DA neurons during PQ induced toxicity and the neuroprotection conferred by GA. The current major goal in PD field is to search for alternative effective therapy and this study is solely aligned with this goal. Overall the work is highly relevant and the authors did utilized different approaches to justify their findings. The work is well written, however I have few minor comments.

Author's comments: We are thankful to the reviewer for the overall positive comments on our manuscript.

1. On the introduction section, since the GA is the focus of this study, I would recommend to introduce it earlier on rather than waiting for the reader to know what is GA only in the discussion part. This can be stated in the 4th paragraph of the introduction where authors are describing plant derived molecules and their use to manage neurodegenerative diseases. One would wish to know which plant GA is isolated from or if it synthetic one. Also if there are any studies on GA that are related to neurodegeneration or neuroprotection would not harm mentioning the key ones in the introductory section.

1.Author's comments: As suggested by the reviewer, we have added a few lines about GA in the Introduction section on Pages 5 and 6 (highlighted text).

2. On the methodology section; while reading the manuscript especially in the result section, I realized that all the different techniques or assays used, the feeding or substance administration to flies was similar or the same. For example the concentration of PQ, the time of exposure, concentration of GA, concentration of sucrose etc. I thought instead of repeating the explanation in each technique or assay used, such information can be once captured in details in the methodology section in the subsection of Drosophila feeding regimen, then in the result section to remain with very specific ones.

2.Author's comments: As suggested by the reviewer, we have tried to rearrange the experimental details to get rid of redundant information from the results section.

3.Authors did not state anywhere in the manuscript why or what was the criteria of choosing four (4) days for GA pre-feeding prior to PQ exposure, why not 2 or 5. If there are reasons, would be good to state them so that it can be clear to the reader, as it is now one remains with a question why four?

3.Author's comments: We thank the reviewer for this comment and have included the justification in the revised manuscript (page 8). The criteria for choosing four days was based on our stringent preliminary screening experiments where we compared the survival data for two, four and five days of pre-feeding and observed that four days of GA prefeeding at 10uM dosage yielded maximum protection that was highly reproducible against PQ-induced toxicity.

4.Authors should check if it is necessary to separate fig. 1c and fig.1e. Both figures are showing negative geotaxis assay and reading on the figure legend of fig 1, authors stated the time points in which the assays were conducted in which both were taken at same time points (24 h and 48 h). I I would suggest to combine the graphs, this will also allows a reader to see the effect of L-DOPA in comparison to GA during PQ exposure in one single graph!

4.Author's comments: We thank the reviewer for the comments. The justification for using two different graphs to show the data was the difference in the mode of feeding. As for GA and GB were active when pre-fed for 4 days prior to PQ exposure. In contrast, L-dopa was protective in co-feeding experiments along with PQ. Therefore, the sucrose fed controls were different in both cases and the reason for not combining the graphs into one. We have transferred the data on L-Dopa to Supplemental Fig 1.

5. On the result section on page 7 under the subsection with a title - effect of structurally related GardeninB.....line 4 from top - a word files should be removed. It seems to be typing error.

5.Author's comments: We thank the reviewer for pointing that out and we have corrected the typing error in the revised manuscript. We have also combined Figs 1 and 2 as suggested by Reviewer#1.

Please do not hesitate to contact us if you have any questions or concerns. We look forward to hearing from you.

Thank you for your time,

Urmila Maitra, Ph.D.

Lukasz Ciesla, Ph.D.

Department of Biological Sciences, Box 870344,

2320 Science and Engineering Complex (SEC),

University of Alabama, Tuscaloosa, AL 35487-0344.

Email: lmciesla@ua.edu; umaitra@ua.edu

Tel: 205-348-1828

REVIEWERS' COMMENTS:

Reviewer #3 (Remarks to the Author):

The authors have addressed all the raised comments which improved their work.